# Human Adenovirus Molecular Characterization in Various Water Environments and Seasonal Impacts in Riyadh, Saudi Arabia

**DOI:** 10.3390/ijerph18094773

**Published:** 2021-04-29

**Authors:** Islam Nour, Atif Hanif, Adel M. Zakri, Ibrahim Al-Ashkar, Abdulkarim Alhetheel, Saleh Eifan

**Affiliations:** 1Botany and Microbiology Department, College of Science, King Saud University, Riyadh 11451, Saudi Arabia; ahchaudhry@ksu.edu.sa; 2Biotechnology Laboratory, Department of Plant Production, College of Food and Agriculture Sciences, King Saud University, Riyadh 11451, Saudi Arabia; azakri@ksu.edu.sa (A.M.Z.); ialashkar@ksu.edu.sa (I.A.-A.); 3Agronomy Department, Faculty of Agriculture, Al-Azhar University, Cairo 11651, Egypt; 4Laboratory Medicine, Department of Pathology, College of Medicine, King Saud University, Riyadh 11451, Saudi Arabia; aalhetheel@ksu.edu.sa

**Keywords:** human adenovirus, waterborne, type 41, prevalence, seasonality

## Abstract

The regular monitoring of water environments is essential for preventing waterborne virus-mediated contamination and mitigating health concerns. We aimed to detect human adenovirus (HAdV) in the Wadi Hanifah (WH) and Wadi Namar (WN) lakes, King Saud University wastewater treatment plant (KSU-WWTP), Manfouha-WWTP, irrigation water (IW), and AnNazim landfill (ANLF) in Riyadh, Saudi Arabia. HAdV hexon sequences were analyzed against 71 HAdV prototypes and investigated for seasonal influence. ANLF had the highest HAdV prevalence (83.3%). Remarkably, the F species of HAdV, especially serotype 41, predominated. Daily temperature ranges (22–45 °C and 10–33 °C) influenced the significance of the differences between the locations. The most significant relationship of ANLF and IW to WH and KSU-WWTP was found at the high-temperature range (*p* = 0.001). Meanwhile, WN was most correlated to ANLF at the low-temperature range (*p* < 0.0001). Seasonal influences on HAdV prevalence were insignificant despite HAdV’s high prevalence in autumn and winter months, favoring low temperatures (high: 22–25 °C, low: 14–17 °C) at five out of six locations. Our study provides insightful information on HAdV prevalence and the circulating strains that can address the knowledge gap in the environmental impacts of viruses and help control viral diseases in public health management.

## 1. Introduction

Surface water resources are usually contaminated with wastewater discharge [1,2]. As a result, there continue to be major public health issues due to the contamination of water resources, such as recreational waters, rivers, as well as wastewater treatment plants (WWTPs), by fecal-mediated enteric viruses [3,4,5]. 

Human adenovirus (HAdV), a commonly detected enteric virus, is a non-enveloped icosahedral virus with a double-stranded linear DNA genome of 34–36 kb [6]. They are currently categorized into seven HAdV species, A to G, in addition to the recently identified adenovirus types that are continuously emerging [7,8]. Although cross-neutralization has defined serotypes up to type 51, genotype description has been mostly implemented for the more recently identified types that reveal the presence of novel sequences or recombinant phylogeny in the HAdV genes encoding the major capsid proteins [9].

Adenoviruses infections can result in asymptomatic cases [10] as well as a wide spectrum of clinical symptoms. Gastrointestinal infections are frequently caused by species A, D, and F. On the other hand, species B is a major cause of infections of the lung and urinary tract. Meanwhile, species C and E are mainly responsible for infections of the respiratory tract. Notably, serotypes 40 and 41 of species F, as well as serotype 31 of species A, are mainly associated with gastroenteritis [11]. Adenoviruses are commonly self-limiting, but not in immuno-deficient individuals. However, a previously emerged adenovirus type 14 strain has caused serious respiratory disease in healthy personnel [12].

HAdV is specific to humans, although adenoviruses can infect a wide range of animals. In domestic sewage, HAdV was found in significantly high concentrations, and their seasonal variability was doubtful [13,14,15]. Like most enteric viruses, adenoviruses survive in the environment and even during sewage treatment processes better than currently used fecal indicator bacteria [16,17]. Moreover, adenoviruses in water environments have been extensively investigated via the sequencing of cloned PCR products or even the direct Sanger sequencing of PCR products [18,19,20,21].

The prevalence of waterborne HAdV in Saudi Arabia has not been extensively studied. Thus, the current study aimed to detect the circulating HAdV in six different locations, that is, in two lakes, two wastewater treatment plants, irrigation water, and a landfill of wastewater, in Riyadh, Saudi Arabia. Moreover, the hexon sequences of the detected HAdV were sequenced for phylogenetic analysis. Furthermore, the seasonal impact was examined in the context of HAdV prevalence in these sampling locations.

## 2. Materials and Methods

### 2.1. Samples Collection

HAdV in various water reservoirs in Riyadh was evaluated. From April 2018 till March 2019, 500-mL samples were collected thrice monthly from each sampling area including two valleys, two wastewater treatment plants (WWTPs), irrigation water (IW) of King Saud University (KSU), and the An-Nazim wastewater landfill. The two valleys were Namar Valley (NV) and Hanifa Valley (HV), while the WWTPs were the Southern Riyadh Wastewater Treatment Plant at Manfouhah (MN-WWTP) and KSU-WWTP. The treated water from KSU-WWTP is used for landscape irrigation and as a coolant at the KSU-based power plant. MN-WWTP effluents are used for flushing sewers as well as controlled irrigation [22,23]. Moreover, the water at HV and the subsidiary NV includes the partially treated and domestic drainage from Riyadh besides usages for recreation [24]. 

### 2.2. Viral Concentration

HAdV was recovered via PEG-mediated protein precipitation according to the procedure of Bibby and Peccia (2013) with some modifications [25]. First, 25 mL of glycine buffer (0.05 M glycine, 0.3 g/L beef extract, pH 9.6) was added to 200 mL of a water sample to detach the virions from the organic matter before centrifugation at 8000× *g* for 30 min. The resultant supernatant was filtered with a 0.2-µm syringe filter (Corning, NY 14831, USA), followed by filtrate treatment with PEG 8000 and NaCl at 80 g/L and 17.5 g/L, respectively, and agitation at 100 rpm under room temperature conditions. The viruses were then obtained by centrifuging at 13,000× *g* for 30 min, removing the supernatant, and centrifuging at 13,000 × *g* for 5 min to obtain a pellet. The virus was eluted from the pellet with 1 mL phosphate buffer saline (PBS) and then stored at −80 °C.

### 2.3. Nucleic Acid Extraction and Specific PCR Detection

HAdV DNA was extracted from 200 µL of a water sample using the PowerViral^®^ Environmental RNA/DNA Isolation Kit (MO BIO, Carlsbad, CA, USA) following the manufacturer’s instructions. HAdV was detected using 2× Phusion Master Mix (Thermo Fisher Scientific, USA) for hexon gene-specific PCR. The 20-μL PCR mixture consisted of a 2-μL DNA template, 300-nM forward primer (AdFhex-F; 5′-GCCACCGATACCTACTTCAGCCTG-3′), 300-nM reverse primer (AdFhex-R; 5′-GGCAGTGCCGGAGTAGGGTTTAAA-3′) [26], and 1× Phusion Master Mix. The PCR was conducted at 98 °C for 30 s, followed by 40 cycles of 98 °C for 10 s, and 72 °C for 30 s, and a final extension at 72 °C for 5 min. A HAdV-positive sample was provided by the Virology Unit of King Khalid University Hospital, Riyadh, Saudi Arabia, to act as the process control for virus extraction and PCR amplification.

### 2.4. Amplicon Purification and Sequencing

Agarose gel electrophoresis was applied at 2× concentration to visualize the PCR products. The resultant 261-bp amplicons were then purified utilizing the Wizard^®^ SV Gel and PCR Clean-Up System (Promega Co., Madison, WI, USA), according to the manufacturer’s protocol. The purified amplicons were sequenced using the ABI genetic analyzer 3130Xl (Applied Biosystems^®^, Carlsbad, CA, USA).

### 2.5. Phylogenetic Analysis

The detected HAdV nucleotide sequences were analyzed using the MEGA X software [27] and compared with the sequences of 71 known HAdV types [10] (Appendix A). The sequence alignments were generated using ClustalW using default settings with an opening penalty of 15 and an extension penalty of 6.66. The phylogenetic tree was constructed according to the best-fit model of nucleotide substitution using the minimum Bayesian information criterion. The reliability of the phylogenetic tree was estimated by the bootstrapping of 1000 replicates. The genetic distances were estimated by the Kimura three-parameter method.

### 2.6. Statistical Analysis

Pearson’s correlation coefficient matrix was implemented to examine the potential relationships between the different sampling areas within a 1-year period. One-way analysis of variance was carried out to test the significance of the impact of the high- and low-temperature ranges on HAdV prevalence regardless of the sampling area. The relationships between different sampling locations, as dependent variables, and high and low temperatures, as independent variables, were fitted using linear curve fitting. All the statistical analyses were conducted using the XL-STAT statistical package software (Ver. 2019, Excel Add-ins soft SARL, New York, NY, USA).

## 3. Results

### 3.1. AnNazim Landfill Exhibited the Highest HAdV Prevalence

Out of the 216 water samples tested for HAdV, 142 (65.74%) were positive for the 261-bp amplicon (Figure 1). The highest HAdV prevalence was found in ANLF at about 83.3%, whereas the lowest prevalence was observed in MN-WWTP at approximately 44.4% (Table 1). The molecular characterization of the amplicon sequences by Sanger sequencing showed 92 sequences; however, the remaining 50 sequences were not recognized because of overlapped electropherograms. 

### 3.2. Predominance of HAdV Serotype 41 (Type F)

The phylogenetic tree illustrated the typical relationship of the HAdV hexon sequences with serotype 41 followed by serotype 40 of HAdV (type F) rather than with the other serotypes (A–E and G) (Figure 2). Pairwise distancing analysis uncovered 13 different HAdV isolates, and most of them (5/13) were collected from ANLF (Appendix A). WH sequences, WH1 and WH2, showed the closest relationship with HAdV type 41 as they belonged to the same branch. However, WH3 was closely related to WN1. Remarkably, ANLF sequences, LF1, LF2, and LF3, had a closer relationship with the sequences from other sampling areas, such as KSU, IW, MN2, and MN, than the WH and WN sequences had, even though they belong to the same branch. Moreover, LF4 and LF5 were closely related to each other. In addition, the notably shorter branch length in the case of detected HAdV sequences indicated a lack of divergence from each other as well as from the closest HAdV serotype 41.

The highest log-likelihood tree is displayed (−2249.38). Heuristic search-dependent initial tree(s) were attained automatically via Neighbor-Join and BioNJ algorithms applied to a pairwise distance matrix that was assessed by the maximum composite likelihood (MCL) approach followed by the highest log likelihood-resulted topology selection. A 5-category discrete Gamma distribution was implemented ([+G]; 0.3272) according to best-fitting substitution model validation to model the difference in the evolutionary rates between sites (Appendix A). The horizontal distance connecting two HAdV sequences was proportional to the genetic distance between them. The distance was expressed as the number of nucleotide substitutions per site. There were a total of 261 positions in the final dataset. The red italicized sequences (Figure 2) denote the current study sequences of accession numbers MW936369 (WH1_HAdV_2018), MW936370 (WH2_HAdV_2018), MW936371 (WH3_HAdV_2019), MW936372 (WN1_HAdV_2018), MW936373 (LF1_HAdV_2018), MW936374 (KSU1_HAdV_2018), MW936375 (IW_HAdV_2018), MW936376 (MN1_HAdV_2018), MW936377 (MN2_HAdV_2019), MW936378 (LF2_HAdV_2018), MW936379 (LF3_HAdV_2018), MW936380 (LF4_HAdV_2018), and MW936381 (LF5_HAdV_2019). 

### 3.3. Relationship Differences in Sampling Areas

The relationships between the sampling areas with respect to HAdV detection at high temperatures (the numbers above the gray highlight, Table 2) differed from those at low temperatures (the numbers below the gray highlight, Table 2). At the high-temperature range, ANLF and IW were of the highest significant correlation (*p* = 0.001) to WH and KSU-WWTP, respectively, followed by a slightly lower correlation of ANLF to WN (*p* = 0.017) and WN and MN-WWTP to WH and KSU-WWTP (*p* = 0.039 and 0.038, respectively) in terms of % HAdV detection. In contrast, at the low-temperature range, WN had the highest correlation to ANLF (*p* < 0.0001), followed by a significant correlation of KSU-WWTP to MN-WWTP, IW, and ANLF (*p* = 0.001, 0.001, and 0.028, respectively), MN-WWTP to IW, ANLF, WH, and WN (*p* = 0.006 and 0.014, 0.024, and 0.04, respectively), WH to WN and ANLF (*p* = 0.012 and 0.014, respectively), and ANLF to IW (*p* = 0.021) (Figure 3).

### 3.4. Non-Significant Influence of Seasonal Variation on HAdV Prevalence

Regardless of the sampling area, HAdV prevalence was not found to be significantly influenced by variations in temperature (*p* = 0.256), despite a notable decline in HAdV prevalence in late summer (August 2018) (Figure 4). Similarly, the segregation of sampling areas displayed no significant influence of high or low temperature on HAdV prevalence (Table 3). However, the highest HAdV prevalence was mostly favored in the lowest-temperature ranges (22–25 °C) at five sampling areas in the high-temperature range, unlike the HAdV prevalence detected at MN-WWTP at the high-temperature range (26–29 °C). Surprisingly, HAdV detection was highest at 14–17 °C in the low-temperature range, equivalent to 26–29 °C at the high-temperature range (Figure 5). 

## 4. Discussion

Human enteric viruses impose critical public health issues owing to their considerable stability in water resources, low infectious doses, and significant concentrations of viral shedding. The accidental release of feces, body fluids, or improperly treated wastewater is a common route of enteric viral contamination in aquatic environments [4,28]. HAdVs have been detected in groundwater, surface water, sewage, and even wastewater treatment plants worldwide [13,29,30,31,32]. 

We detected HAdV in all the investigated water resources, consistent with previous studies. This observation could be due to HAdV prevalence, human cell tropism, or their resilience in various environments and populations [18,33]. Therefore, HAdV may be an index indicator of wastewater fecal contamination [34].

On the other hand, there are limited studies on HAdV prevalence in the water resources in Saudi Arabia since most studies focused on detecting circulating HAdV strains in patients [35,36,37,38]. The current study focused on HAdV prevalence in various water environments. The highest HAdV prevalence was observed in wastewater landfills, likely as a result of the frequent, extensive disposal of biosolids in landfills that increase the level of pathogens in the resultant leachate [39]. Moreover, landfills and associated facilities were reported to have a HAdV prevalence of 75% [40], less than that reported in the current study at 83.3%. The higher HAdV prevalence is due to the landfills in Saudi Arabia receiving dewatered sludge from most WWTPs as solid waste that lacks stabilization, thus disseminating enteric viruses and contaminating shallow groundwater aquifers [41]. 

The present study also investigated the serotypes of the circulating HAdV strains in water. Notably, all the detected HAdV strains belonged to species F, usually associated with acute gastroenteritis in children [18]. A previous study on a membrane bioreactor-based wastewater treatment plant in Saudi Arabia detected two types of HAdV, species F and species A [41]. These observations are partially consistent with our results. Moreover, HAdV serotypes 40 and 41 were detected in diarrheal patients from Riyadh, Jeddah, and Mecca in Saudi Arabia [36], in line with our observations that our detected HAdV sequences cluster with serotype 41. 

In addition, varied relationships were observed between the sampling areas. ANLF and IW were of the most significant relationship to WH and KSU-WWTP at a high-temperature range, as KSU-WWTP is the primary source of IW, besides the waste disposal of WH to ANLF. Likewise, WN was highly correlated to ANLF at the low-temperature range because of the potentially linked waste disposal routes. In addition, WN, as a subsidiary branch of WH, contributed to a significant relationship between the lakes [24]. 

Moreover, higher HAdV prevalence was detected at lower temperatures in autumn and winter, in agreement with previous findings [42,43]. However, Wang et al. [44] recorded higher HAdV prevalence in summer and spring than autumn, similar to a previous study conducted in China that reported a higher prevalence in summer [45]. The discrepancy may be due to geographical differences, the lack of significance observed in the relationship between temperature and HAdV prevalence in Wang et al. [32], or the difference in HAdV types detected in China encompassing HAdV type 3 and type 7 as the most prevalent types. Meanwhile, our current study detected HAdV type 41. Despite the higher HAdV prevalence in autumn and winter, seasonal variation showed no significant influence on HAdV prevalence. The lack of seasonal tendency of HAdV was previously reported in several studies [14,15].

## 5. Conclusions

This study provides insightful information concerning HAdV prevalence in various water environments in Riyadh, uncovers the predominance of the species F of HAdV serotype 41 in Saudi Arabia over other HAdV species, and reports a lack of significance of seasonality influence on HAdV prevalence. However, continuous monitoring of circulating HAdV in Saudi Arabia during notable weather changes is essential to address potential public health concerns, especially in cold months due to the corresponding higher prevalence of HAdV.

## Figures and Tables

**Figure 1 ijerph-18-04773-f001:**
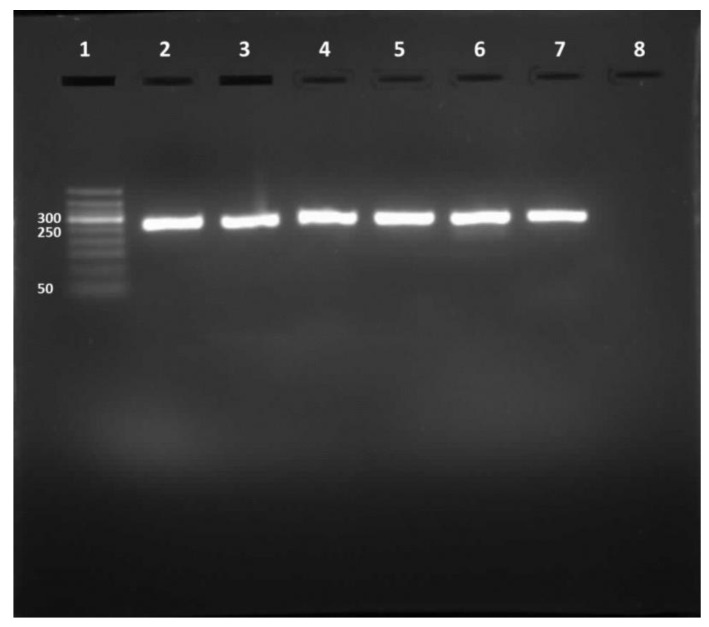
RT-PCR product. Lane 1, DNA ladder (50–500 bp). Lane 2, HAdV positive control. Lanes 3 to 7, 261-bp HAdV amplicons. Lane 8, negative control.

**Figure 2 ijerph-18-04773-f002:**
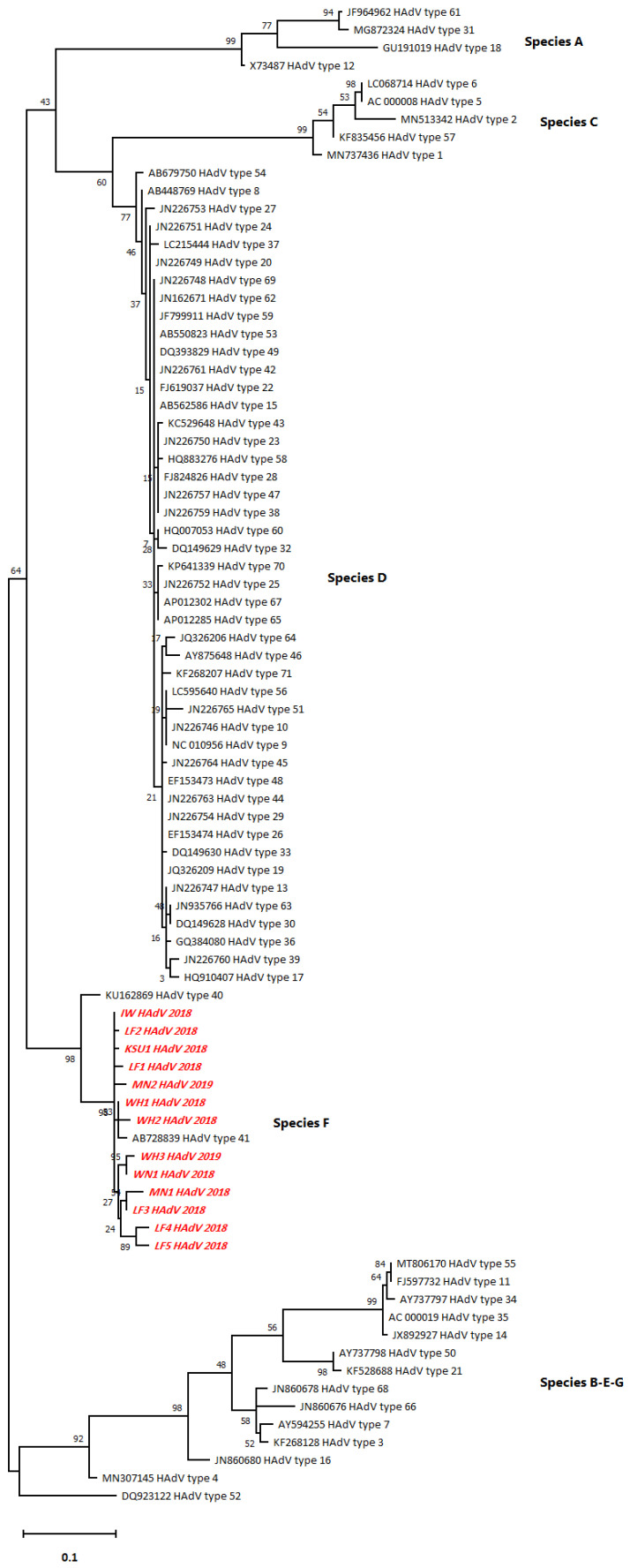
The phylogenetic tree for the HAdV hexon sequences constructed by the maximum likelihood method and Tamura 3-parameter model.

**Figure 3 ijerph-18-04773-f003:**
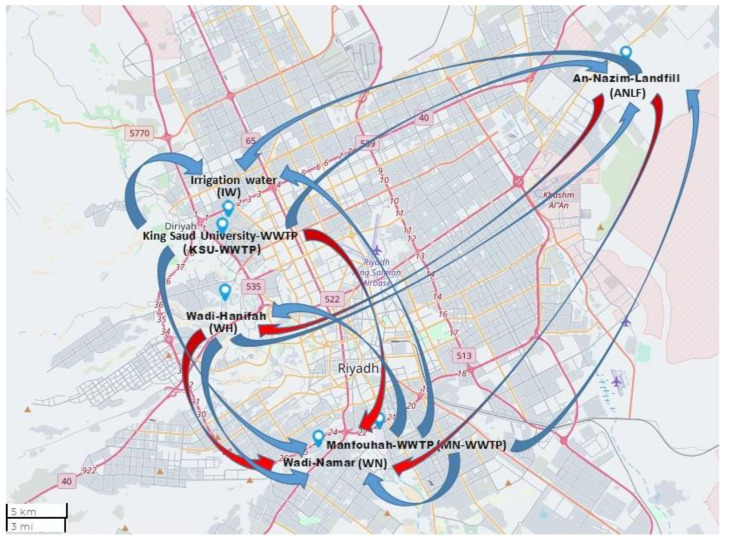
Map showing correlations between sampling areas at the higher temperature range (red arrows) and lower temperature range (blue arrows), using Mapline integrated Excel Addin 2016 (Mapline Co., Provo, UT 84604, USA).

**Figure 4 ijerph-18-04773-f004:**
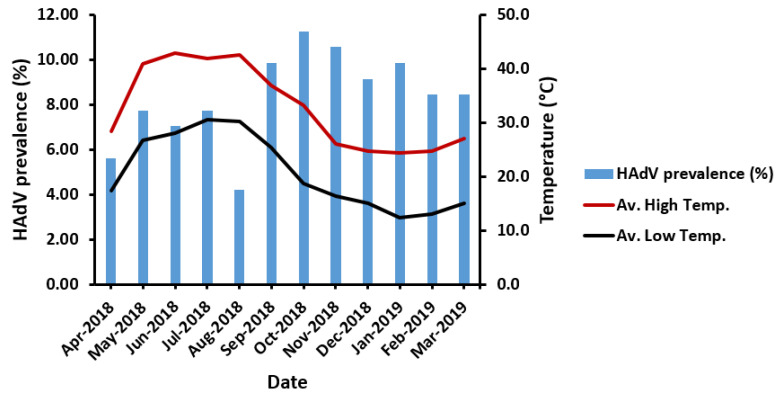
The impact of temperature variation on HAdV prevalence. Av. High Temp., the average high temperature; Av. Low Temp., the average low temperature.

**Figure 5 ijerph-18-04773-f005:**
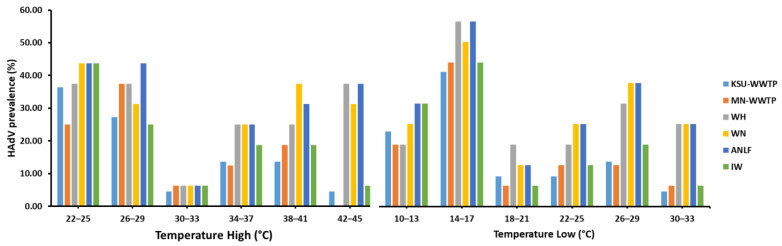
The influence of temperature variations on HAdV prevalence in different sampling locations.

**Table 1 ijerph-18-04773-t001:** HAdV cases in different sample locations.

Sampling Area	HAdV Cases in 2018 (April–December)	HAdV Cases in 2019 (January–March)	HAdV Prevalence (%)
KSU-WWTP	15	7	61.11
MN-WWTP	11	5	44.44
WH	21	6	75.0
WN	22	6	77.78
ANLF	22	8	83.33
IW	13	6	52.78

KSU-WWTP, King Saud University Wastewater Treatment Plan; MN-WWTP, Manfouhah-WWTP; WH, Wadi Hanifah; WN, Wadi Namar; ANLF, AnNazim landfill; IW, irrigation water.

**Table 2 ijerph-18-04773-t002:** Pearson’s correlation matrix of HAdV detection percentage at the various sampling areas.

	% Det_KSU-WWTP_	% Det_MN-WWTP_	% Det_WH_	% Det_WN_	% Det_ANLF_	% Det_IW_
**% Det_KSU-WWTP_**		**0.836**	0.578	0.667	0.688	**0.971**
**% Det_MN-WWTP_**	**0.976**		0.422	0.463	0.568	0.720
**% Det_WH_**	0.811	**0.871**		**0.833**	**0.977**	0.523
**% Det_WN_**	0.767	**0.832**	**0.908**		**0.892**	0.710
**% Det_ANLF_**	**0.861**	**0.901**	**0.901**	**0.982**		0.630
**% Det_IW_**	**0.971**	**0.937**	0.728	0.780	**0.880**	

The numbers above the gray-highlighted diagonal refer to correlation values at the high-temperature range, whereas those below the diagonal are correlation values at the low-temperature range. The percentage of HAdV detectability is denoted by “% Det” at different sampling areas. For example, % Det_KSU-WWTP_ refers to the HAdV detection percentage at KSU-WWTP. Significant correlation values are displayed as bold numbers.

**Table 3 ijerph-18-04773-t003:** The significance of the influence of high or low-temperature ranges on HAdV prevalence in various sampling areas.

Sampling Area	Temperature Range	R^2^	RMSE	Equation
KSU-WWTP	High	0.641	8.544	% Prev_KSU-WWTP_ = 58.94 − 1.36 × T_H_ ^‡^
	Low	0.476	10.829	% Prev_KSU-WWTP_ = 40.11 − 1.23 × T_L_
MN-WWTP	High	0.480	10.885	% Prev_MN-WWTP_ = 55.42 − 1.25 × T_H_
	Low	0.325	12.924	% Prev_MN-WWTP_ = 30.24 − 0.71 × T_L_
WH	High	0.007	13.754	% Prev_WH_ = 32.28 − 0.13 × T_H_
	Low	0.025	16.178	% Prev_WH_ = 34.06 − 0.31 × T_L_
WN	High	0.011	14.356	% Prev_WN_ = 34.70 − 0.18 × T_H_
	Low	0.011	14.356	% Prev_WN_ = 32.56 − 0.18 × T_L_
ANLF	High	0.035	15.652	% Prev_ANLF_ = 42.32 − 0.36 × T_H_
	Low	0.073	15.917	% Prev_ANLF_ = 41.43 − 0.54 × T_L_
IW	High	0.553	10.414	% Prev_IW_ = 62.69 − 1.38 × T_H_
	Low	0.476	12.145	% Prev_IW_ = 46.09 − 1.38 × T_L_

% Prev denotes HAdV prevalence percentage at different locations. RMSE refers to the root mean squared error, which is an absolute measure of fit. ^‡^ T_H_ represents the highest temperature, whereas T_L_ denotes the lowest temperature.

## Data Availability

The sequences used in this study for phylogenetic analysis are openly available in NCBI GenBank repository with accession numbers prementioned in Figure 2.

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
