# Peer review of "Human Adenovirus Molecular Characterization in Various Water Environments and Seasonal Impacts in Riyadh, Saudi Arabia"

_ijerph, 2021, doi:10.3390/ijerph18094773_

Round 1

Reviewer 1 Report

Nour et al. in their paper “Human Adenovirus molecular characterization in various water environments and seasonal impacts in Riyadh, Saudi Arabia” aimed to detect human adenovirus (HAdv) in a different location in Saudi Arabia. Their results showed that AnNazim had the highest virus population and the temperature degree affected the virus population. Their study is important because it reveals the prevalence of HAdV in Saudi Arabia and how the environmental factor (temperature changes) could affect that.

The authors discussed in detail the importance of studying the virus in the introduction section, the health effect of that as well as the different isolates and serotypes. The material and methods section is well described and gives in-depth information about each analysis step (with minor issues in section 2.5):

  1. They did not mention which MEGA version did they use
  2.  Besides, they did not explain why did they use 6.66 as an extension penalty for sequence alignment. 
  3. Also, it would be interesting to explain based on what did they choose 71 HAdV isolates and from which country (If possible)?

With that said, I have some question:

  1. Did the authors start with the same number of samples for each region?
  2. The number of cases dropped between 2018 and 2019, the authors did not suggest any reason.
  3. The authors showed a difference in virus concentration related to temperature, is there any variant clustered concerning temperature change?
  4. Did the sequenced 92 samples were submitted to the NCBI database? (I did not see any accession number regarding them)

Minor comments:

  1. Table S2 should be in CSV, Tab, or Excel sheet rather than in word.

Author Response

Dear Reviewer,

I highly appreciate your highly valuable comments and for offering time for reviewing our paper. I have made some modifications as per your guidance accompanied with annotated reply in a separate file.

Kindly, find the annotated reply file entitled "Reviewer comments and authors' replies" attached.

Best regards,

Authors 

Reviewer 2 Report

This paper deals with the occurrence of adenoviruses in various sources of water in Saudi Arabia. The results are of limited interest and the study raises several important concerns:

Table one: it seems that different numbers of samples were collected from the various sources.   Together only 260 samples were collected. It is a small number to use as the basis for statistical calculations. Only 142 were positive for adenovirus DNA by PCR analysis. How was the cut off level defined?

It seems that collections were made in 2 years, 2018 and 2019. The number of positive samples differs vastly between the two years. Collected in different seasons??

Sanger sequencing was used to study the PCR amplicons. The results revealed 92 sequences but not all were readable. What is the meaning of overlapped electropherograms?

The phylogenetic tree depicted in Fig 2 is based on comparisons of DNA sequences, which were only 261 bp long.

The extensive statistical analysis is difficult to follow and does not give many meaningful insights

The figure legends are incomplete and very difficult to follow. In fig 2, I  assume that the red text indicates samples identified in the study

Author Response

(The authors gave the same response as above.)

Reviewer 3 Report

This article studied the molecular characterization of Adenovirus in various environments in Riyadh. The water samples were collected from different sites and analyzed for the presence of Human Adenovirus. AnNazim landfill exhibited the highest HAdV prevalence, with a predominance of serotype 41. The relationship between the different sampling areas highlighted a links between these sites and depended of the temperature. In contrast, the HAdV prevalence was not influence by the seasonal variations.

The study is well conducted and interesting. Nevertheless, improvements are required:

  1. A figure who represents the links between the different sample sites can help the reader to understand and interpret the table 2. A map could also be useful.
  2. A quantitative Real time PCR would allow estimation of the “viral load” in each sample, and a comparison between each site.
  3. Idem for seasonal variation. Is there the same quantity of HAdV depending of the temperature?

Author Response

(The authors gave the same response as above.)

Round 2

Reviewer 1 Report

Nour et al. in their paper “Human Adenovirus molecular characterization in various water environments and seasonal impacts in Riyadh, Saudi Arabia” aimed to detect human adenovirus (HAdv) in a different location in Saudi Arabia. Their results showed that AnNazim had the highest virus population and the temperature degree affected the virus population. Their study is important because it reveals the prevalence of HAdV in Saudi Arabia and how the environmental factor (temperature changes) could affect that. 

Thank you for your work.

Reviewer 2 Report

The text is improved.

Reviewer 3 Report

Accept in present form.